# On Ausferrite Produced in Thin Sections: Stability Assessment through Round and Flat Tensile Specimen Testing

Giuliano Angella [1,*], Riccardo Donnini [1], Dario Ripamonti [1], Franco Bonollo [2], Bogdan Cygan [3] and Marcin Gorny [4]

1. Research Institute CNR-ICMATE, DSCTM, Via R. Cozzi 53, 20125 Milano, Italy
2. Department of Management and Engineering, University of Padua, Str. S. Nicola, 3, 36100 Vicenza, Italy
3. Teksid Iron Poland, Ciężarowa 49, 43-430 Skoczów, Poland
4. Faculty of Foundry Engineering, AGH University of Science and Technology, Władysława Reymonta 23, 30-059 Kraków, Poland
* Correspondence: giuliano.angella@cnr.it

**Abstract:** Ductile irons were produced into different casting wall sections, that is, 25 mm, 5 mm and 3 mm. The alloys were then austenitized with the same conditions at 875 °C for 2 h and austempered for three different combinations of temperatures and times: 250 °C for 6 h, 310 °C for 3 h and 380 °C for 1 h. The aim of the investigation was to study the ausferrite stability of austempered ductile irons with three different nominal contents of nickel produced in thin sections through tensile testing. So, strain hardening analysis of tensile flow curves was carried out since it has been found to be a reliable support to ductility analysis in assessing the optimal austempering conditions. Because of different wall sections, round and flat tensile specimens with geometries complying with ASTM E8/E8M-11 were tested. Austempered ductile irons from 5 and 3 mm wall sections were tested through flat geometry specimens only, while 25 mm wall sections were tested through both round and flat geometries. Though the ausferrite was affected by Ni content and the graphite morphology was improved with reduced thin sections, the ausferrite stability and the tensile mechanical behavior were insensitive to Ni content and section thickness below 25 mm. Furthermore, it resulted that the tensile plastic behavior was sensitive to the specimen geometry in a consistent way, increasing the instability of ausferrite and indicating that a proper analysis and comparison of tensile properties of austempered ductile irons must take into account the tensile specimen geometry.

**Keywords:** austempered ductile iron; ausferrite stability assessment; strain hardening; specimen geometry

## 1. Introduction

Austempered ductile irons (ADIs) are advanced spheroidal irons produced through heat-treating conventional ductile irons (DIs), resulting in a dual phase acicular microstructure called ausferrite, consisting of hard bainitic ferrite $\alpha$ and metastable high C content austenite $\gamma_{HC}$ [1–8]. Ausferrite has an excellent combination of strength, ductility [9–13] and other mechanical properties, such as fatigue and fracture resistance [14–23]. Their properties are similar to cast and wrought steels, with which they can compete as materials for applications in components for heavy transportation, such as trucks, earth-moving machinery and the train industry [24–27], thanks also to their production cost that is about 50% less than the cost of steels and their density that is about 10–12% lower [7,28,29].

The ADI production route consists of two-step heat treatments: a conventional DI is first austenitized at high temperature, typically at about 900 °C, to have homogeneous austenite rich in C; then it is quenched in a salt bath to maximize the heat transfer and is subsequently held at a constant temperature, typically between 250 °C and 380 °C, to trigger the austempering solid-state transformation [1–6]. After a proper austempering

time, the resulting microstructure is the dual phase acicular ausferrite, that is, $\gamma_{HC} + \alpha$, and finally, the system is cooled down slowly to room temperature to avoid any residual stress. However, if the system is held at the austempering temperature for longer times, the reaction $\gamma_{HC} \rightarrow \alpha + \varepsilon$ occurs, where $\varepsilon$ is a FeC carbide that dramatically reduces the ausferrite ductility. So, for the production of optimal ausferrite, the proper austempering time window has to be found to maximize the austempering reaction $\gamma \rightarrow \alpha + \gamma_{HC}$ and avoid the detrimental reaction $\gamma_{HC} \rightarrow \alpha + \varepsilon$.

The $\gamma_{HC}$ and $\alpha$ volume fractions, the acicular microstructure dimension and the $\alpha$ hardness because of upper or lower bainite transformation depend mainly on the austempering temperature [1–6]. Higher austempering temperatures, such as 380 °C, produce coarse acicular or feathery ausferrite with higher volume fractions of $\gamma_{HC}$ coming into lower yield and tensile strengths, and better ductility. Lower austempering temperatures, such as 250 °C, produce fine ausferrite with lower volume fractions of $\gamma_{HC}$, resulting in higher yield and tensile strengths, and reduced ductility. However, it has also been reported that austenitization temperature affects ausferrite since lower austenitization temperatures increase the driving force for austempering reaction, raising the reaction rate and the stability of the resulting ausferrite [6,30–33]. Furthermore, since acicular ferrite formation starts at interfaces of graphite/austenite and inclusions or austenite boundaries, the resulting austempering kinetics are accelerated by fine original microstructure [32,33].

For the quality assessment of ausferrite, the maximum volume fraction of austenite is not the only parameter to be controlled since the higher the C content in austenite, the more stable austenite is. In fact, with austempering, the martensite-start temperature (Ms) should be suppressed at 0 K so that the C-rich austenite is thermally stable at room temperature. Indeed, the C-rich austenite may be neither thermally nor mechanically stable at low temperatures but transforms into martensite as a consequence of cryogenic treatments and external loading [8,24,25,34,35], reporting that stress-induced and deformation-induced austenite-to-martensite ($\gamma \rightarrow M$) transformation occurs in ADIs, which could contribute to the high ausferrite hardness and tensile strength [24,25]. The evaluation of the tensile mechanical properties of ADIs has to be carried out to assess definitely the stability of ausferrite and, as a consequence, to evaluate the goodness of the austempering production parameters [12,13,36–38]. Tensile ductility is usually analyzed to find the optimal austempering time, but the variability in tensile ductility is indeed the major problem in this approach; on the contrary, the analysis of tensile strain hardening through the dislocation-density-related constitutive equation, such as the Voce equation, has been proved to be very good for optimal austempering time assessment [12,13]. A *Matrix Assessment Diagram* (MAD) is built up by plotting the Voce equation parameters ($1/\varepsilon_c$ vs. $\Theta_o$) found by fitting the tensile strain hardening data of ADI tensile flow curves with the Voce equation. In MADs, data from a single austempered heat that has been heat-treated with a single austempering temperature and different austempering times can be fitted with a best-fitting line, and the Voce data positions in MADs identify the best austempering time [12]. This method can also be used for comparison between ADIs with different chemical compositions that have been heat-treated with the same austempering conditions to identify which chemical composition better matches the imposed austempering setting [13].

Graphite–metallic matrix boundaries affect ferrite nucleation during austempering, so different nodule counts and sizes have effects on the austempering kinetics, which has been modeled in [32] and experimentally measured with dilatometry investigations in [33]. In ADIs, Cu, Ni and Mo are added to improve austemperability [39–42], that is, to avoid stable pearlite formation and to foster homogeneous ausferrite formation. However, increasing alloying results in potentially increasing chemical segregations with heterogeneous final ausferrite, which can be significant in thicker sections, causing the degradation of ADIs' mechanical properties [43–45]. So, thin sections that have a higher nodule count and chemically homogeneous metallic matrix should result in a more homogeneous ausferrite with a little blocky austenite in which martensite might form after loading or cryogenic application. To the authors' knowledge, none of the literature has reported on the effects

of different thin sections on ausferrite stability. In this work, the results of the investigations on the effects of the solidification rates (thickness sections) of the original DIs on the resulting ADIs' microstructure and tensile mechanical properties are reported. Castings with different wall thicknesses, namely 25 mm, 5 mm and 3 mm, with different Ni contents were austenitized at the same temperature and time, and then were austempered at three different temperatures and time conditions. Because of the different ADI sections, tensile specimens with round and flat geometries were tested to assess the tensile mechanical properties and the strain-hardening behavior of the different sections. So, the side investigation was the assessment of specimen geometry on the plastic behavior and stability of ausferrite. The microstructures of the ADI thin sections were analyzed using Scanning Electron Microscopy (SEM) and X-ray Diffraction (XRD).

## 2. Materials and Experimental

### 2.1. Original Cast Irons and Austempering Conditions

The original DIs were produced in Teksid Iron Poland, Skoczów, Poland, through different nominal contents of nickel (0.0, 0.7 and 1.5 wt%) into different casting wall sections: 25 mm, 5 mm and 3 mm. Castings with a wall thickness of 25 mm were produced complying with Y-blocks ASTM A536–84(2019)e1 [46], while castings with wall thicknesses of 5 and 3 mm were taken from step test castings. Experimental melts were carried out in medium-frequency induction furnaces with a capacity of 12 tons. The standard procedure for the metal charge was applied: 60% circulating own scrap and 40% steel scrap, with carburizer to correct the content of C. After melting the charge and reaching a temperature of 1420 °C, the slag was removed from the surface of the liquid cast iron, and control samples were taken to assess the chemical composition with emission spectrometry and the metallurgical quality of liquid cast iron. Experimental molds in green sand technology for making castings for testing the microstructure and mechanical properties were made. In mold technology for spheroidization and inoculation, processes were carried out using automatic pouring devices. The tests of C content were made with the LECO apparatus from Mg content using the Atomic Absorption Spectrometry (AAS) method. In Table 1, the chemical compositions of the three different heats are reported.

**Table 1.** Chemical compositions of the three ADI heats in wt%.

| Code | C | Si | Mn | S | P | Mg | Cu | Ni |
|---|---|---|---|---|---|---|---|---|
| 0.0 wt% Ni | 3.500 | 2.554 | 0.326 | 0.015 | 0.046 | 0.035 | 0.762 | 0.012 |
| 0.7 wt% Ni | 3.520 | 2.624 | 0.305 | 0.013 | 0.036 | 0.050 | 0.736 | 0.736 |
| 1.5 wt% Ni | 3.520 | 2.540 | 0.357 | 0.012 | 0.042 | 0.037 | 0.700 | 1.533 |

The alloys were then austenitized at 875 °C for 2 h, and austempered for three different combinations of temperatures and times: 250 °C for 6 h; 310 °C for 3 h; 380 °C for 1 h. Tensile testing was performed with round tensile specimens with geometries complying with ASTM E8/E8M-11 [47], with a strain rate of $10^{-4}$ 1/s and strain control up to rupture. In Figure 1, the Ultimate Tensile Strengths (UTS) in MPa and Elongations to Rupture ($e_R$) in % resulted from the average of four tensile tests on ADIs produced with different Ni contents, and 25 mm Y-blocks are reported together with the minimum properties required according to ISO17804:2005 [48]. Indications of the austempering temperatures are reported on the plot for clarity's sake. The produced ADIs were all complying with [48], with UTS and elongations to rupture higher by far than those required. Indeed, only the 1.5 wt% Ni ADI austempered at 380 °C for 1 h matched the minimum properties 900-8; however, elongations to rupture were shown to be lower by far than those of the ADIs with 0.0 and 0.75 wt% Ni austempered at the same austempering temperature.

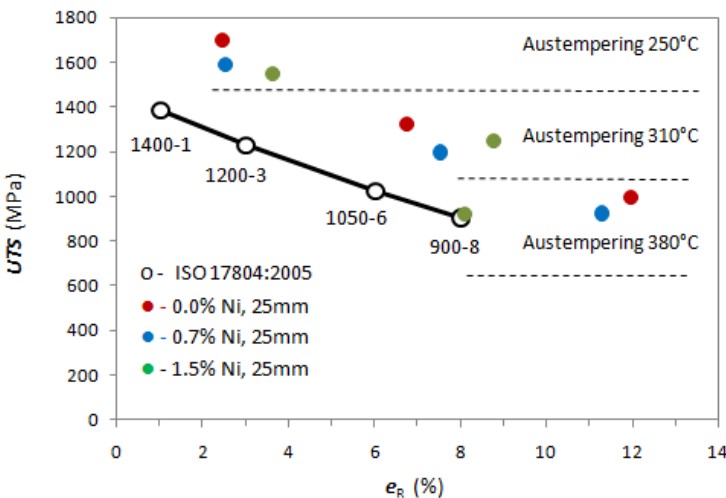

**Figure 1.** UTS (MPa) vs. $e_R$ (%) for the ADIs produced with different Ni contents and with three different austempering temperatures (austenitization similar for all compositions at 875 °C for 2 h) with round tensile specimens from 25 mm Y-blocks. Minimum tensile properties complying with ISO17804:2005 [48] are reported for comparison purposes. Indications of the austempering temperatures are reported on the plot for clarity's sake.

## 2.2. Microstructure Analysis

The microstructure was observed in Scanning Electron Microscopy (SEM) with the microscope SU70 by Hitachi, after conventional grinding and polishing, and Backscattered Electron Imaging (BEI) for graphite morphology observations with polishing and then chemical etching with 2% Nital and Secondary Electron Imaging (SEI) for ausferrite observations. Nodularity and nodule count were calculated through digital image analysis complying with ASTM E2567-16a [49], working on BEI micrographs because the reduced gray scale typical of BEI made easier the binarization for image processing.

In order to quantify the volume fractions of retained austenite ($V_\gamma$) and to relate them to the observed tensile properties, X-ray Diffraction (XRD) analysis was performed for each sample on properly ground and polished surfaces by conventional metallographic methods. An X-ray diffractometer Siemens D500 with Bragg-Brentano geometry and Cu Kα radiation ($\lambda$ = 0.1542 nm) was used, and XRD patterns were collected in the 2$\theta$ range 20–110° with a step size of 0.02° and 5 s of dwell time. Volume fractions of austenite and ferrite were calculated complying with the ASTM E975-13 [50]; corresponding intensities and peak 2$\theta$ positions were obtained for the identified $(111)_\gamma$, $(110)_\alpha$, $(200)_\gamma$, $(200)_\alpha$, $(211)_\alpha$, $(311)_\gamma$ and $(220)_\alpha$ lattice planes by peak approximation with Pearson type VII distribution [51]. For the graphite volume fraction ($V_\gamma$), the calculated value of 8% by the aforesaid image analysis was considered. The C content in $\gamma$ austenite was evaluated according to the empirical equation [52]

$$C^\gamma_{s\,\{111\}} = \left(a^\gamma_x - 0.3573\right)/0.0033 \tag{1}$$

where $a_x{}^\gamma$ is the lattice parameter of $\gamma$ austenite cell defined as

$$a^\gamma_x = \left(\frac{\lambda}{sin2\theta}\right)^2 \frac{1}{4}\left(h^2 + k^2 + l^2\right) \tag{2}$$

and $h$, $k$ and $l$ are the Miller indices of the crystallographic planes.

## 2.3. Tensile Testing and Flow Curves Analysis

Because of the different casting sections, round and flat tensile specimens with geometries complying with [47] were tested in strain control with strain rate $10^{-4}$ 1/s. ADIs from 5 and 3 mm wall sections were tested through flat geometry specimens only, while 25 mm wall sections were tested through both geometries. In Figure 2, the picture of flat and round

specimens tested are reported. The gauge ridges were used to connect extensometers and control gauge elongation up to rupture. In the engineering tensile data, stress $S = F/A_o$ and strain $e = (l - l_o)/l_o$, where $F$ is the applied load, $A_o$ and $l_o$ are the initial cross-section and length of the tensile gauge, while $l$ is the instantaneous length of the tensile gauge, were converted into true stress $\sigma$ vs. true strain $\varepsilon$ data according to the relationships $\sigma = S \cdot (1 - e)$ and $\varepsilon = \ln(1 + e)$. The plastic component ($\varepsilon_P$) only of strain was used, that is, $\varepsilon_P = \varepsilon - \sigma/E$, where $E$ is the experimental Young modulus.

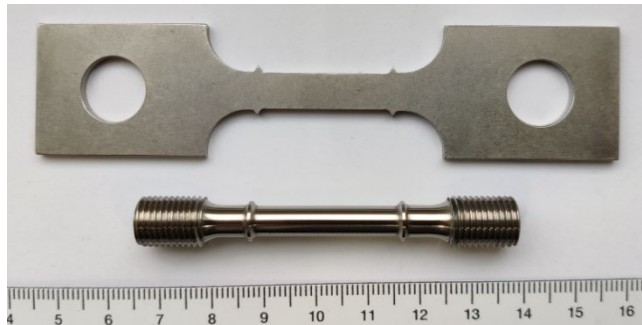

**Figure 2.** Flat and round geometries of the tensile specimens (marker unit in cm).

As reported in [53–57], the procedure to find out the Voce equation parameters is based on the analysis of the differential data $d\sigma/d\varepsilon_P$ vs. $\sigma$, where $d\sigma/d\varepsilon_P$ is the strain hardening rate. According to the Kocks–Mecking model of strain hardening [57–59], strain hardening rate and stress are linearly related:

$$\frac{d\sigma}{d\varepsilon_P} = \Theta_o - \frac{\sigma}{\varepsilon_c} \qquad (3)$$

where $\Theta_o$ and $1/\varepsilon_c$ are constants with physical meaning that describe the micro-mechanics of plastic deformation and are related to the matrix microstructure [57]. An example of strain hardening analysis based on the Voce equation is reported in Figure 3 for 1.5 wt% Ni ADI from 25 mm Y-block austempered at 380 °C for 1 h.

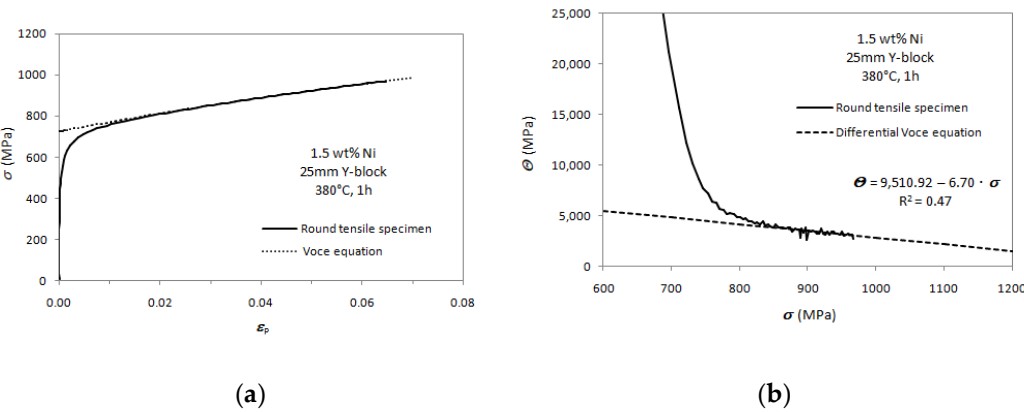

(**a**)          (**b**)

**Figure 3.** Tensile plastic behavior fitting with Voce equation: (**a**) tensile flow curve with round tensile specimen of 1.5 wt% Ni ADI from 25 mm Y-block austempered at 380 °C for 1 h; (**b**) differential data from the flow curve in a) and strain hardening analysis based on Voce equation with best linear fit: intercept $\Theta_o$ = 9510.92 MPa; slope $1/\varepsilon_c$ = 6.70.

It has been reported [12,13,53–57] that through plotting $1/\varepsilon_c$ vs. $\Theta_o$ of a statistically meaningful set of flow curves, the *Matrix Assessment Diagram* (MAD) is built up with which the cast iron can be classified, and its quality can be assessed. When Voce tensile data from an ADI produced through different austempering times are plotted in MAD, an evaluation of the optimal time for austempering reaction can be made [12,13].

For each 25 mm Y-block, four tensile round specimens were tested for the assessment of the minimum tensile properties reported in Figure 1. The flow curves from round specimens were also used to build up the MAD for all different chemical compositions and austempering conditions. From the thin sections (5 and 3 mm), only flat tensile specimens could be machined off and tested. So, in order to compare properly the different sections, also from 25 mm Y-blocks, flat tensile specimens were tested to evaluate any possible specimen geometry effect, if existing.

## 3. Results

### 3.1. Microstructure Characterization

The 25 mm Y-block cast iron microstructure produced with the slowest solidification rate presented the smallest nodule count and the biggest average nodule size while, with decreasing thickness, the section nodule count increased and the average nodule size decreased. This graphite trend was the same for all heats. Nodularity and nodule count vs. thickness with different chemical compositions are reported for comparison in Figure 4. The solidification rate had a significant effect on the nodule count in Figure 4a since the nodule count was around 300 mm$^2$ in 25 mm castings, between around 500 and 700 mm$^2$ in 5 mm castings and around 900 mm$^2$ in 3 mm castings. On the contrary, the nodularity was excellent for all sections, as it was always around and higher than 90%, so well above the threshold reported in ASTM A247-19 [60] for good nodularity, i.e., 80%. Finally Ni content did not seem to affect the nodule count and nodularity in the 25 mm, 5 mm and 3 mm castings.

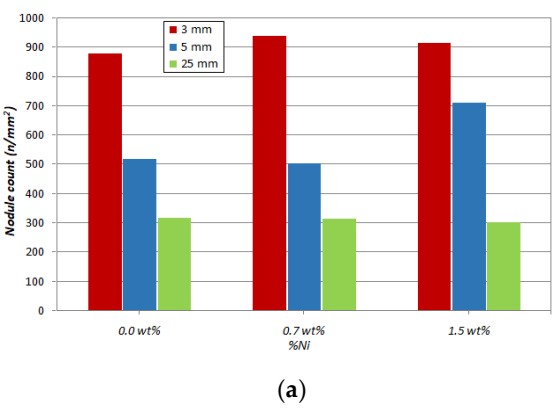

(**a**)

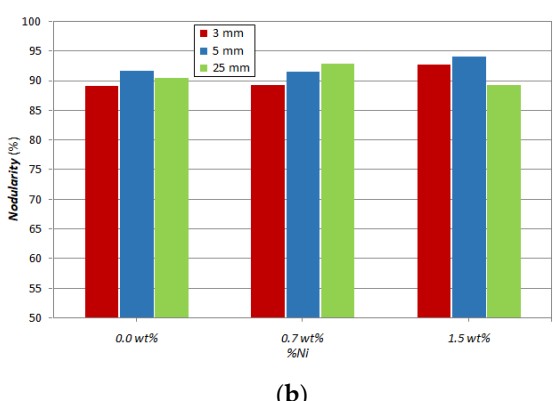

(**b**)

**Figure 4.** Graphite microstructure parameters vs. thickness and Ni content: (**a**) nodule count (number of nodules/mm$^2$); (**b**) nodularity (%).

### 3.2. Ausferrite Characterization

Ausferrite was observed through SEM, and it was thinner after the lowest austempering temperatures of 250 °C, increasing in size with increasing austempering temperatures at 310 °C and 380 °C. Selected SEM images of ausferrite in ADI with 1.5 wt% of Ni after austempering at 380 °C for 1 h from 25 mm, 5 mm and 3 mm wall thickness castings are reported in Figure 5, for instance. Though ausferrite appeared to be organized in smaller packets in the 3 mm casting (Figure 5c) and increased in packets in 5 mm and 25 mm casting ausferrite in Figure 5b and Figure 5a, respectively, ausferrite appeared to be similar, equally coarse and feathery. The packet size seemed to reflect the original grain size that was smaller in the 3 mm casting, but no influence of the nodule count on the ausferrite morphology and size seemed to be evident. Similar findings were gathered for the other Ni contents' ausferrite. The ausferrite morphology was not significantly affected by Ni alloying, resulting in similar at the same austempering conditions with different Ni contents. In Figure 6, selected micrographs of ausferrite with different Ni content produced at the same austempering conditions (310 °C for 3 h) and thickness (25 mm Y-blocks) are reported, for instance.

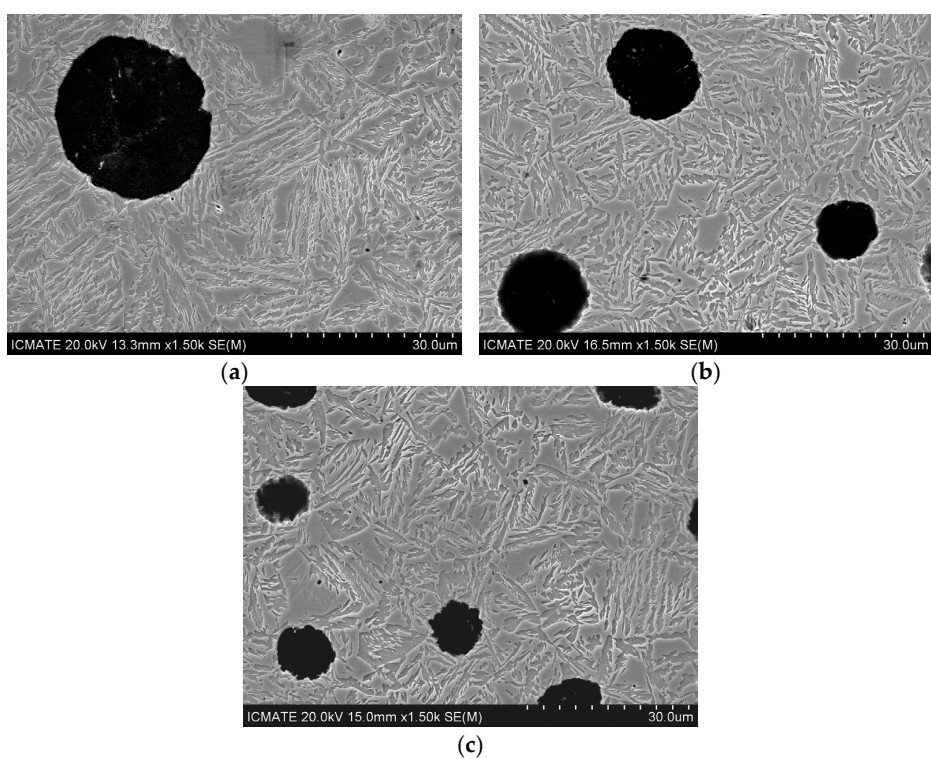

**Figure 5.** Selected SEM micrographs through Secondary Electron Imaging (SEI) of ADIs castings with 1.5 wt% of Ni after austempering at 380 °C for 1 h: (**a**) 25 mm wall thickness; (**b**) 5 mm wall thickness and (**c**) 3 mm wall thickness casting.

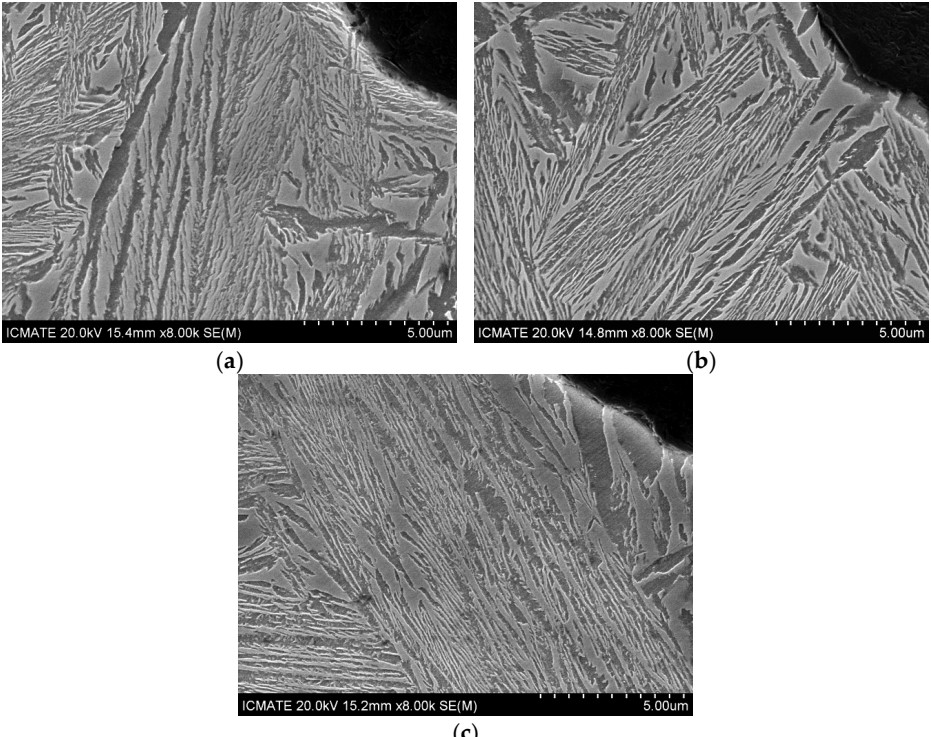

**Figure 6.** Selected SEM micrographs through SEI of ADIs castings with different Ni contents after austempering at 310 °C for 3 h: (**a**) 0.0 wt% 25 mm Y-blocks; (**b**) 0.7 wt% 25 mm Y-blocks and (**c**) 1.5 wt% 25 mm Y-blocks.

Fracture surfaces from tensile specimens were observed through SEM. In Figure 7, micrographs of selected fracture surfaces from round and flat tensile specimens of ADI with 0.0 wt% of Ni produced in 25 mm Y-blocks after austempering at 310 °C for 3 h are reported, for instance. The fracture surfaces presented generally ductile features and few brittle cleavage planes, and no differences in the fracture aspects between round and flat tensile specimens of the same ausferrite were found.

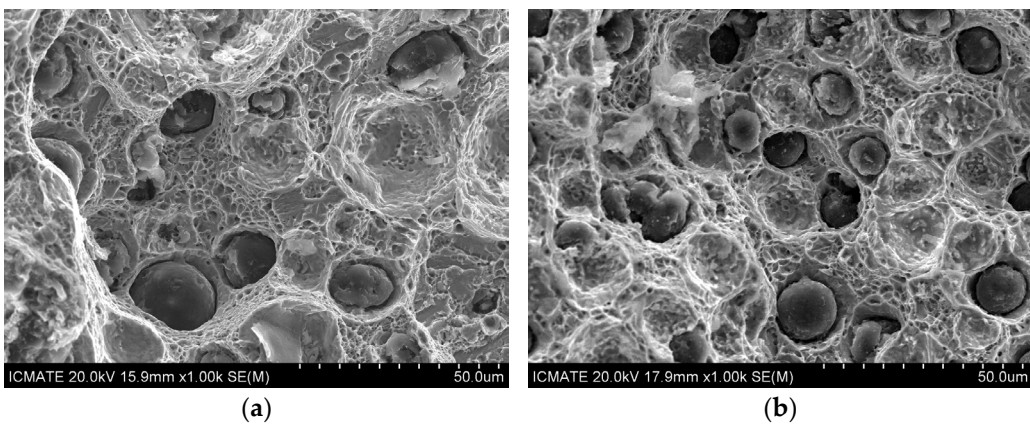

(**a**)　　　　　　　　(**b**)

**Figure 7.** Selected SEM micrographs through SEI of tensile specimen fracture surfaces of ADI with 0.0 wt% of Ni produced in 25 mm Y-blocks after austempering at 310 °C for 3 h: (**a**) fracture surface from round tensile specimen; (**b**) fracture surface from flat tensile specimen.

Examples of representative XRD diffraction patterns of ausferrite (normalized to the ferrite peak $\alpha(110)$) after different austempering conditions and corresponding lattice plane identification are reported in Figure 8 for 1.5 wt% of Ni ADI produced in 25 mm Y-blocks. With increasing austempering temperature, the relative intensity of austenite peaks (indicated as $\gamma(hkl)$) increased significantly, indicating that the austenite volume fractions increased with higher austempering temperatures. Austenite volume fractions vs. austempering temperature are reported in Figure 9 for all chemical compositions and austempering conditions.

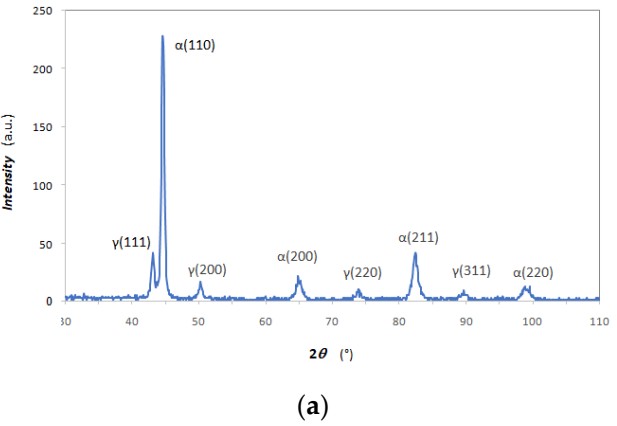

(**a**)

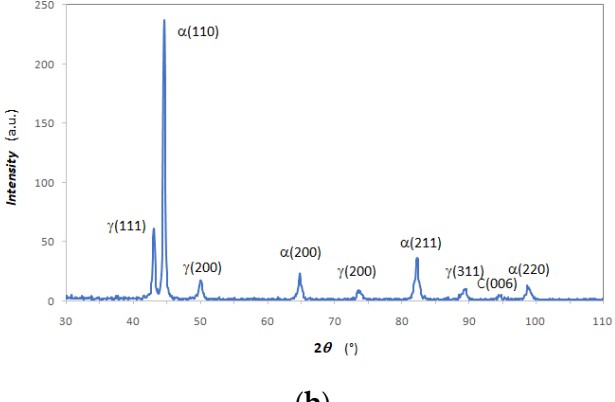

(**b**)

**Figure 8.** *Cont.*

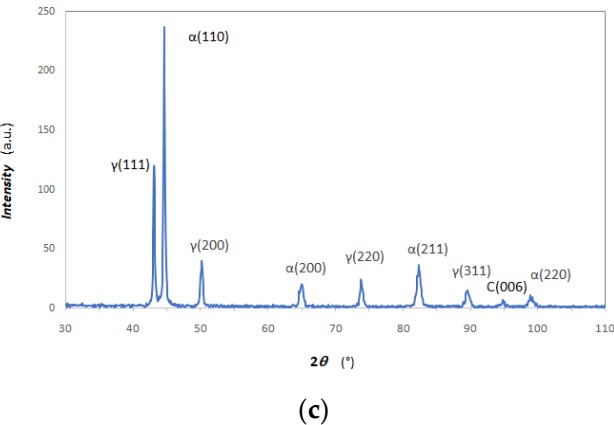

**(c)**

**Figure 8.** Typical XRD patterns of ausferrite (1.5 wt% of Ni, 25 mm Y-blocks) after different austempering conditions: (**a**) 250 °C with 6 h; (**b**) 310 °C for 3 h and (**c**) 380 °C for 1 h.

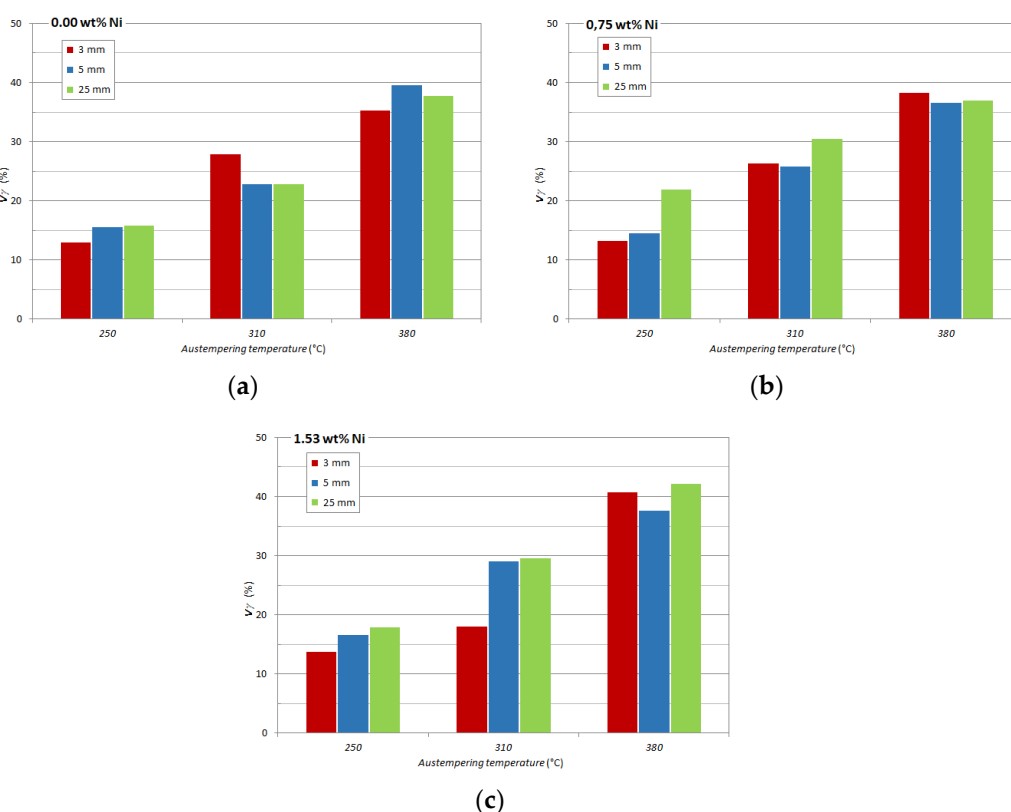

**(a)**                                      **(b)**

**(c)**

**Figure 9.** Austenite volume fraction $V\gamma$ vs. austempering temperature (250 °C for 6 h, 310 °C for 3 h and 380 °C for 1 h); (**a**) 0.00 wt% of Ni; (**b**) 0.75 wt% of Ni; (**c**) 1.53 wt% of Ni.

For all compositions, there was an increase of austenite volume fraction $V\gamma$ with increasing austempering temperature, according to [1–6]. Though no clear effect of wall thickness was evident, with increasing Ni content, the austenite $V_{\gamma}$ seemed indeed to increase. In Figure 10, the average $V_{\gamma}$ of different wall thicknesses with constant Ni content is reported at different austempering temperatures. The Ni-bearing ADIs generally presented an austenite volume fraction higher than the Ni-free ADIs, indicating that Ni fostered the austenite retention, which has been already reported [39,41,42].

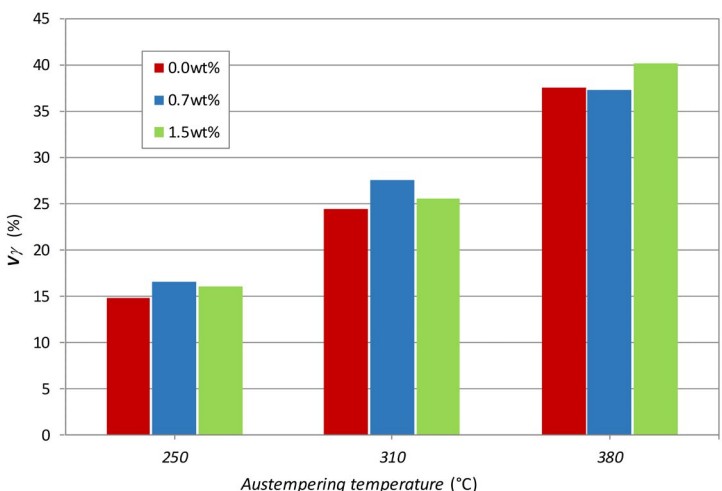

**Figure 10.** Average austenite volume fraction $V\gamma$ vs. austempering temperature (250 °C for 6 h, 310 °C for 3 h and 380 °C for 1 h).

In Figure 11, the $C^{\gamma}{}_S$ values averaged on the different wall thicknesses with constant Ni content are reported at different austempering temperatures. $C^{\gamma}{}_S$ did not seem to be affected by the Ni content and wall thicknesses, while $C^{\gamma}{}_S$ was affected by austempering; the ADIs austempered at 250 °C for 6 h presented the lowest C contents in austenite compared to the values found after austempering at 310 and 380 °C.

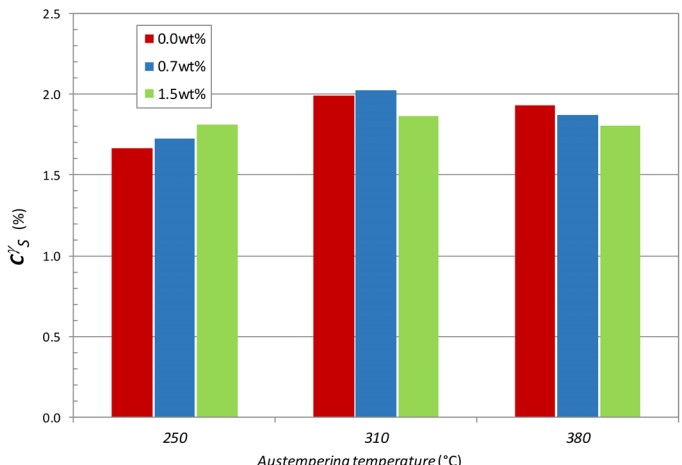

**Figure 11.** Average C content in $\gamma$ ($C^{\gamma}{}_S$) vs. austempering temperature (250 °C for 6 h, 310 °C for 3 h and 380 °C for 1 h).

### 3.3. Tensile Strain Hardening Results on 25 mm Y-blocks: MAD Analysis

Differential data from tensile flow curves of ADIs from 25 mm Y-blocks were analyzed according to the procedure reported in Section 2.3, and the Voce parameters $1/\varepsilon_c$ and $\Theta_o$ were plotted according to the MAD reported in Figure 12. MADs for the three austempering conditions (250 °C for 6 h; 310 for 3 h; 380 °C for 1 h) are reported for the three nominal Ni contents (0.0, 0.7 and 1.5 wt%). In each MAD, the data from each chemical composition comes from four tensile tests carried out on round tensile specimens: the scattering of the data points are indications of ausferrite stability [12,13]. The lower the positions of the data points, the more stable the ausferrite is. So, in Figure 12a, reporting data for ADIs austempered at 380 °C for 1 h, the 0.0 wt% Ni ausferrite was more stable than ADIs with Ni alloying, ranging $1/\varepsilon_c$ from 1.99 to 2.90 (or $\Theta_o$ from 4722.3 to 5697.5 MPa). Because increasing alloying elements content makes the austempering transformation more sluggish, the MAD results suggested that 1 h time was not enough for stable ausferrite

formation with Ni alloying [13], and longer austempering times should have been imposed. In Figure 12b, reporting data for ADIs austempered at 310 °C for 3 h, the 1.5 wt% Ni ausferrite was more stable than ADIs with other compositions, ranging $1/\varepsilon_c$ from 5.82 to 8.30 (or $\Theta_o$ from 10,199.2 to 13,354.9 MPa). In this case, the MAD suggested that 3 h time was too long for stable ausferrite formation with 0.0 and 0.7 wt% of Ni [13], and shorter austempering times should have been imposed, particularly for the 0.0 wt% Ni ausferrite. Finally, in Figure 12c, data from austempering at 250 °C for 6 h are reported; again, the 1.5 wt% Ni ausferrite was more stable than ADIs with other compositions, ranging $1/\varepsilon_c$ from 58.86 to 96.40 (or $\Theta_o$ from 98,585.8 to 157,124.5 MPa). Indeed, the ADI with a nickel content of 0.0 wt% seemed to have a similar behavior to 0.7 wt% Ni content with high Voce values, suggesting that both compositions were far from stable. However, generally, the Voce parameters for all compositions were very high after any austempering conditions, indicating that ausferrite instability was high after austempering at 250 °C.

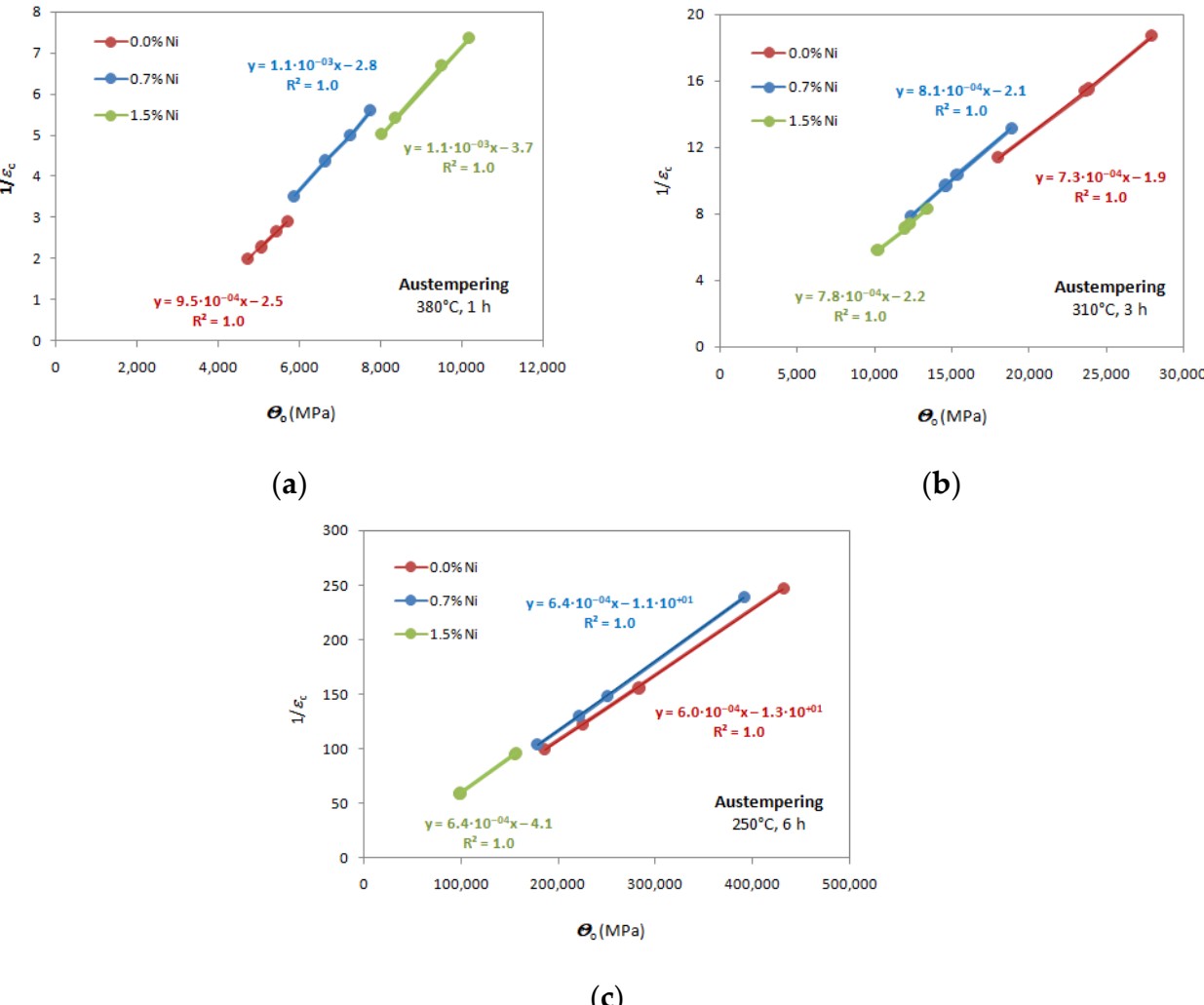

**Figure 12.** *Matrix Assessment Diagrams* (MAD) for round tensile specimens from 25 mm Y-block ADIs of three nominal Ni contents (0.0, 0.7 and 1.5 wt%) and different austempering conditions: (**a**) 380 °C for 1 h; (**b**) 310 °C for 3 h; (**c**) 250 °C for 6 h.

*3.4. Tensile Behavior Comparison between ADIs Produced through 25, 5 and 3 mm Thickness Sections*

Because of the lack of material, only flat tensile specimens could be machined off from the 5 and 3 mm castings. So, for proper comparison flat tensile specimens were also machined off from the ADI 25 mm Y-blocks and then tensile tested. The Voce parameters

were found according to the usual procedure in Section 2.3, and the results are in the MADs reported in Figures 13–15. Additionally, the Voce parameters obtained from the round specimens machined off from the 25 mm Y-blocks (see Figure 12) are reported for comparison purposes, as well as the best linear fits of the round tensile specimens data.

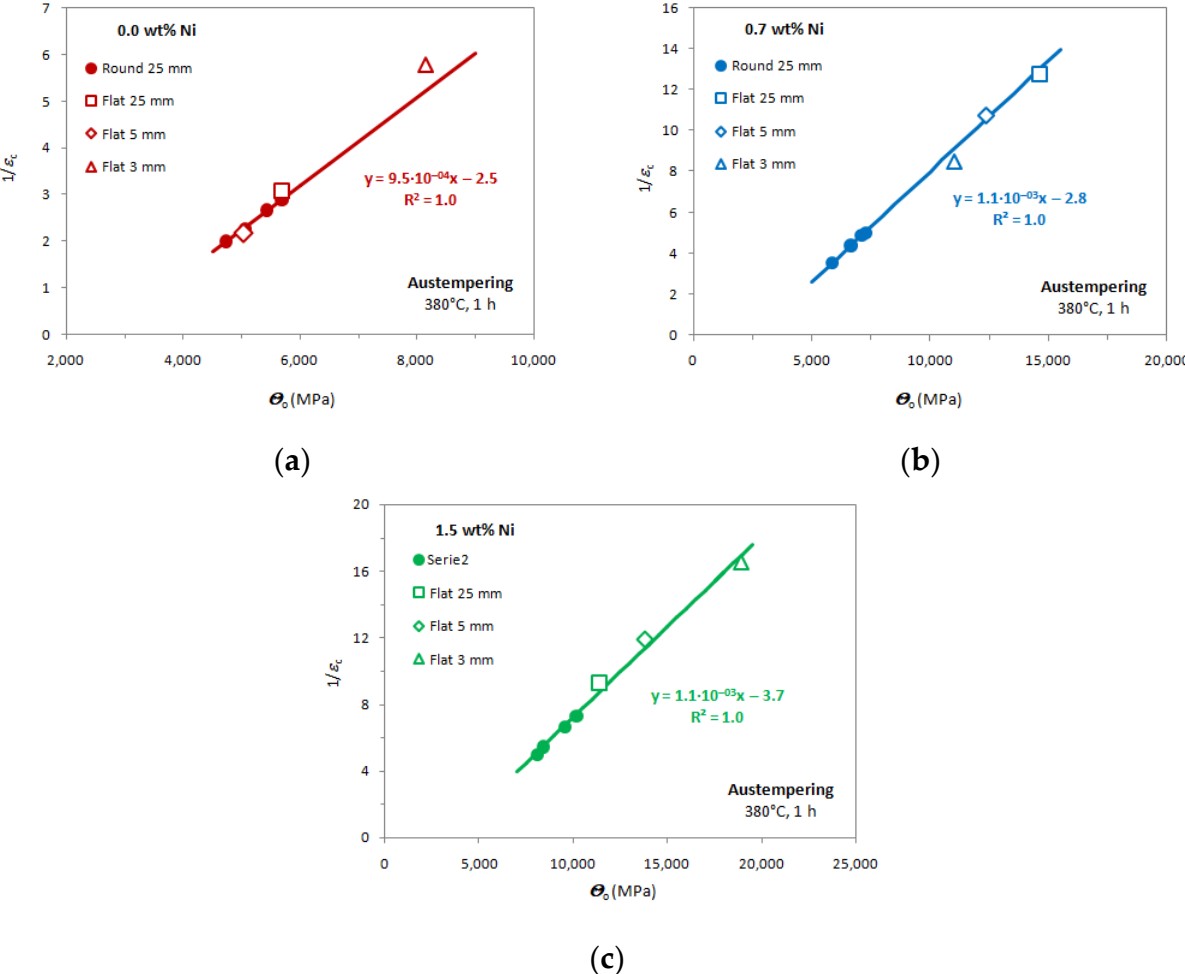

**Figure 13.** MADs for 25mm Y-block ADIs (round and flat tensile specimens), 5 mm and 3 mm thin castings with austempering conditions 380 °C and 1 h with different Ni content: (**a**) nominal 0.0 wt% Ni; (**b**) nominal 0.7 wt% Ni; (**c**) nominal 1.5 wt% Ni.

In Figure 13, the data from austempering at 380 °C for 1 h are reported for the ADIs with the three different Ni contents. It is noteworthy that, in every composition, the data from the 3 and 5 mm flat tensile specimens lie on the best linear fits obtained by the round specimens data of the ADI 25 mm Y-blocks. So, since the Voce parameters from round and flat tensile specimens had the same best linear fits, this finding indicated that the ausferrite from different sections had similar microstructure with possible different stability. So, chemical composition was the relevant parameter for ausferrite formation, regardless of section thickness. However, the Voce data positions of the flat tensile specimens along the best-fitting lines were higher than the round tensile specimens data with apparently worse ausferrite stability. Furthermore, the Voce data position from flat specimens from different thicknesses was also random, which resulted in none of the three wall thicknesses (3, 5 and 25 mm) seeming to have better ausferrite stability. Another interesting result is that the range width of the flat tensile specimens Voce data was by far wider than the Voce data range from round tensile specimens, which would have suggested that ausferrite stability was lower in 5 and 3 mm thin sections. Indeed, the data values from flat specimens of

the 25 mm Y-blocks were also always bigger than the round specimen data from 25 mm Y-blocks, which could indicate that the tensile specimen geometry might have played some role in the tensile flow behavior of ADIs.

In Figure 14, the data from austempering at 310 °C for 3 h are reported for the three different Ni contents. In Figure 14a, the flat tensile specimen data point from 0.0 wt% Ni ausferrite is missing because of the premature rupture, so the short plastic deformation range could not allow for strain hardening analysis. Again, the data from the 3 and 5 mm sections' flat specimens lie on the best linear fits obtained by the round tensile specimens data of the 25 mm Y-blocks. Additionally, for these austempering conditions, the flat tensile specimens data positions along the best-fitting lines were generally higher and random, indicating that no ausferrite of the three sections (3, 5 and 25 mm) had better stability. It is noteworthy that, though the range width of the flat tensile specimens Voce data appeared to be wider by far than the Voce data from round specimens, the data values from flat specimens of the 25 mm Y-blocks were indeed always bigger than the round specimens data from 25 mm sections.

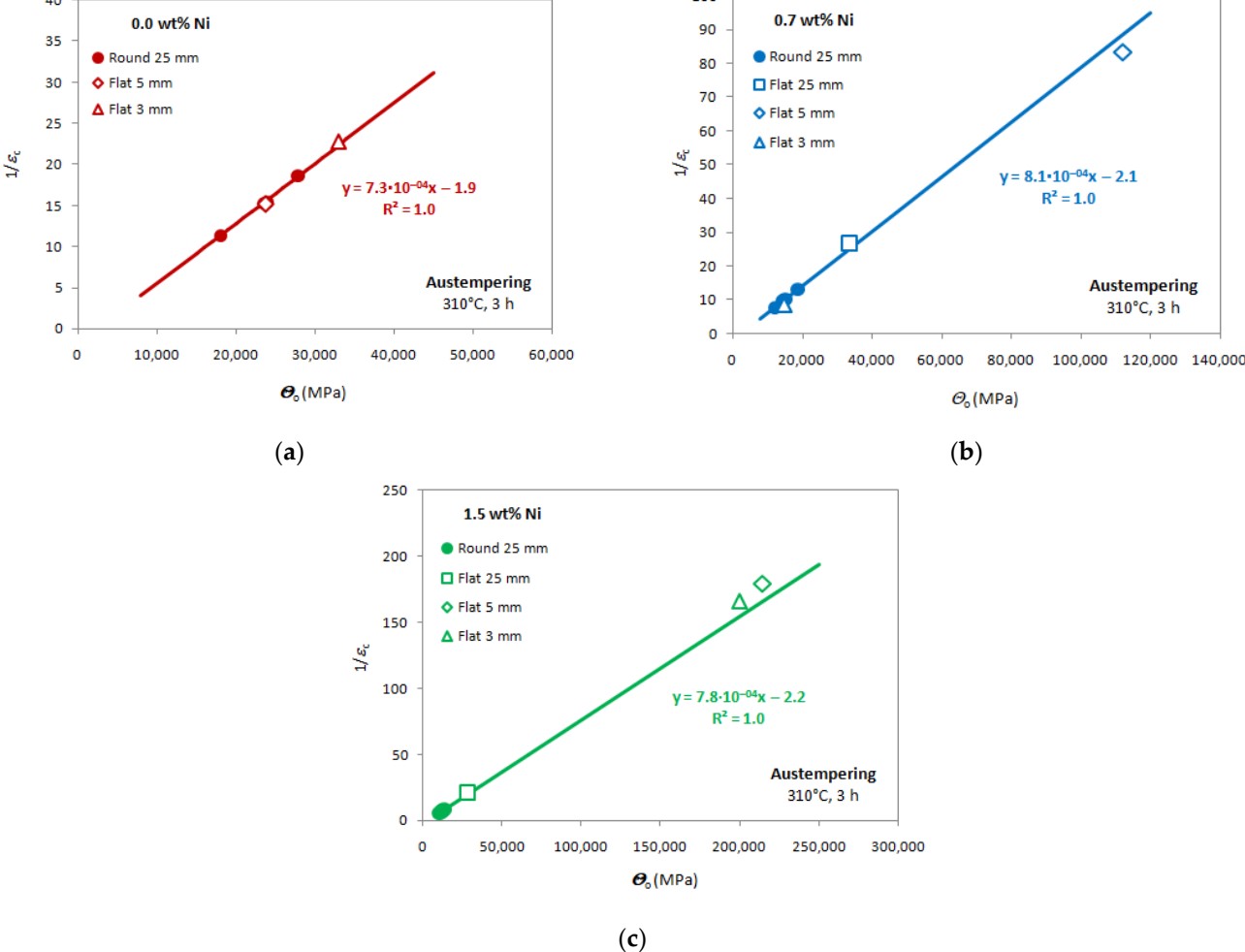

**Figure 14.** MADs for 25 mm Y-block ADIs (round and tensile specimens), 5mm and 3mm thin castings with austempering conditions 310 °C and 3 h with different Ni content: (**a**) nominal 0.0 wt% Ni; (**b**) nominal 0.7 wt% Ni; (**c**) nominal 1.5 wt% Ni.

In Figure 15, the data from austempering at 250 °C for 6 h are reported for the three different Ni contents. The instability of ausferrite after these austempering conditions was generally quite high, causing premature ruptures and short ductility. Even if the 3 mm casting data points were always the lowest regardless of the Ni contents, it was difficult

to state any trend. In fact, the variability of Voce parameters was so high that premature ruptures occurred just after the proof stress, and strain hardening analysis was difficult. That was the reason why some data points are missing in the MADs of austempering at 250 °C in Figure 15.

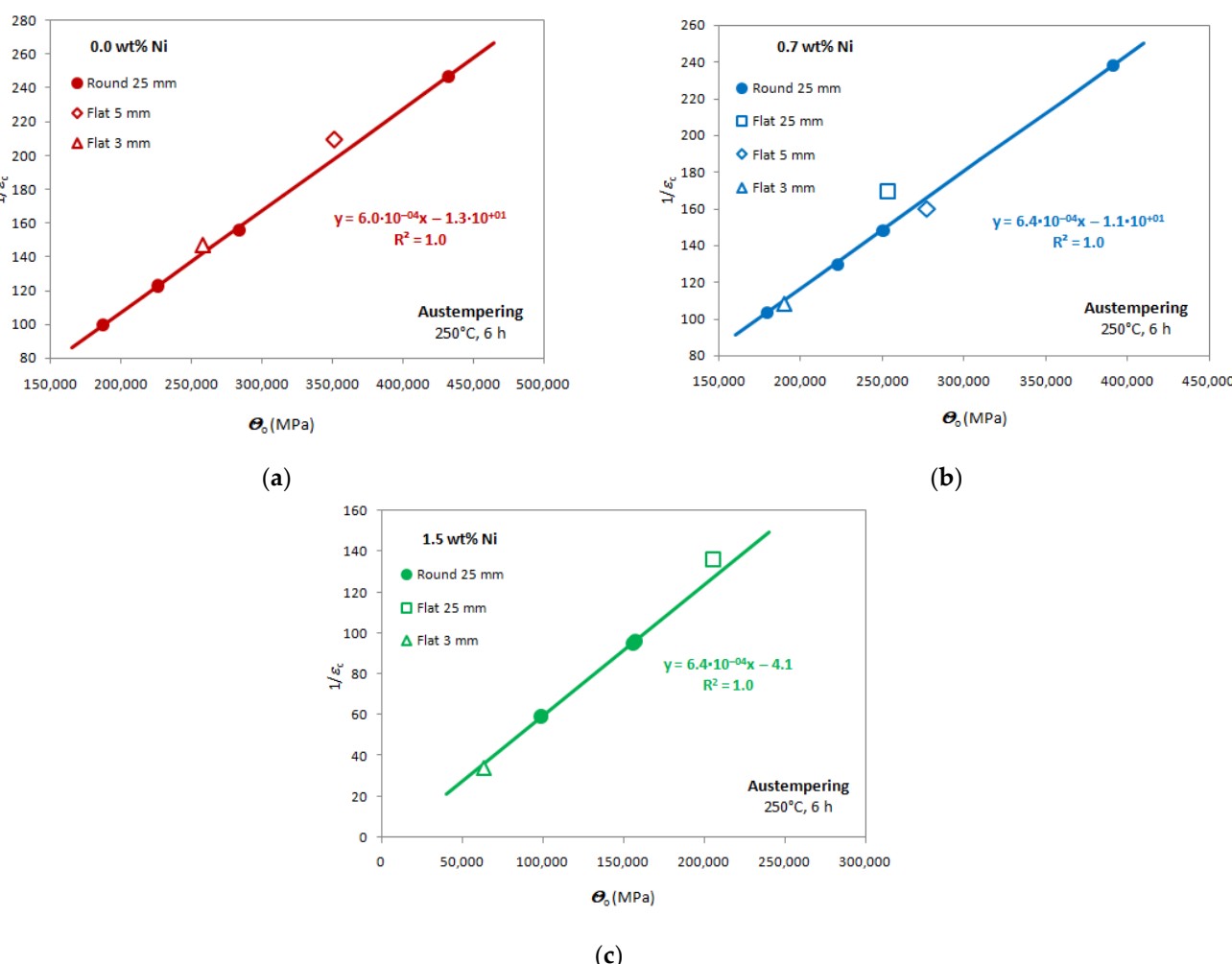

**Figure 15.** MADs for 25 mm Y-block ADIs (round tensile specimens and flat specimens), 5 mm and 3 mm thin castings with austempering conditions 250 °C and 6 h with different Ni content: (**a**) nominal 0.0 wt% Ni; (**b**) nominal 0.7 wt% Ni; (**c**) nominal 1.5 wt% Ni.

## 4. Discussion

### 4.1. Microstructure

The graphite morphology of ADIs was analyzed through digital image analysis complying with the international standard [49]. The nodularity was excellent in all compositions and sections, showing values always above 88% in Figure 4b, so well over the threshold of 80% reported in ASTM 427-19 [60] for high-quality nodular cast irons. The nodule count was instead significantly affected by the section thickness, since nodule count was an average value of about 900 mm$^{-2}$ with the fastest cooling rates in the 3 mm castings regardless of the Ni content, decreasing to an average of about 300 mm$^{-2}$ in the 25 mm Y-blocks. At the intermediate thickness of 5 mm, some variability was found, with a nodule count ranging from about 500 to about 700 mm$^{-2}$ in the 1.5 wt% Ni ausferrite. However, besides these findings, no evidence of Ni influence on the nodule count could be claimed. So, the graphite morphology appeared excellent for any Ni content and consistent with the different cooling rates that seemed to be the only relevant parameter affecting the graphite morphology.

The ausferrite morphology did not seem to be affected by nodule count (section thickness) or Ni content. As reported in Figure 5, the ausferrite seemed to be organized in packets reflecting the original austenite grain size of the material after austenitization. Indeed, mean grain size ($D$) is proportional to half the inter-nodule distance ($\lambda$) through the Fullman's relationship [61]:

$$\lambda = (1 - V_\gamma)/(d \cdot N_G) \qquad (4)$$

where $d$ is the mean nodule size (mm) and $N_G$ is the nodule count (number of nodules/mm$^2$). Through applying equation 4 for 1.5 wt% ausferrite obtained after austempering at 380 °C for 1 h (see Figure 5), $\lambda/2$ resulted in 53.4 µm for 3 mm casting, 60.7 µm for 5 mm casting and 93.5 µm for 25 mm Y-block, which is consistent with the ausferrite packets reported in Figure 5.

The austenite volume fractions $V_\gamma$ vs. austempering temperatures for any Ni content and cooling rate have been reported in Figure 9a–c. With increasing austempering temperature, $V_\gamma$ increased as expected [1–6], passing from an average of about 15% at 250 °C of austempering to about 25% at 310 °C and about 37% at 380 °C. However, with increasing Ni content, the austenite $V_\gamma$ slightly increased, since in Figure 10, where the average $V_\gamma$ of different wall thicknesses with constant Ni content is reported at different austempering temperatures, the Ni-bearing ausferrite generally presented an austenite volume fraction higher than the Ni-free ADIs, indicating that Ni promoted the austenite preservation, as already reported [39,41,42].

Another important parameter to be measured in ausferrite is the C content in metastable austenite: the higher the C content, the more stable the austenite is. In the literature, the value of 1.8 wt% of C is identified as the target to be achieved for good austenite stability [52]. In Figure 11, the carbon content in austenite values ($C^\gamma_S$) averaged for the different wall thicknesses with constant Ni content are reported at different austempering temperatures. $C^\gamma_S$ did not seem to be affected by the Ni content and wall thicknesses, so there was no correlation between metastable austenite volume fractions $V_\gamma$ (that seemed to be affected by Ni content) and C content in austenite. Indeed, $C^\gamma_S$ was affected by austempering: the ADIs austempered at 250 °C for 6 h presented the lowest C contents in austenite below 1.8%, while the values found after austempering at 310 °C and 380 °C were always over 1.8%.

*4.2. Tensile Behavior of 25 mm Y-Blocks with Round Tensile Specimens: Ausferrite Stability through Analysis of MAD and Tensile Mechanical Properties*

The tensile behavior of the ADIs produced in different walls with variable Ni content was investigated according to conventional engineering properties, i.e., UTS and elongations in Figure 1 and according to the *Matrix Assessment Diagram* (MAD) in Figure 10. In the plot UTS vs. elongations to rupture reported in Figure 1, the ADI data points are well above the minimum tensile properties complying with ISO 17804:2005 [48], proving that the austempering conditions selected in this work with differential dilatometry technique were correct and the produced ADIs were excellent. Indeed, the only composition of 1.5 wt% Ni after austempering at 380 °C for 1 h appeared to match the minimum tensile properties [48], but well below the properties measured in the other investigated ADIs.

To give an insight into this finding, the MAD approach based on the Voce analysis of strain hardening was used. In Figure 10, the Voce parameters worked out from the strain hardening analysis of the 25 mm Y-blocks tensile flow curves were reported for any Ni content and austempering condition. So, since different chemical compositions affected the kinetics of the austempering solid reaction, in each MAD the data positions and data point spans were different, indicating that the stability of ausferrite was different after the imposed austempering conditions. In Figure 10a, the austempering conditions of 380 °C for 1 h produced optimal ausferrite for the Ni-free ADI, while alloying with Ni, 380 °C for 1 h was not enough for making the most stable C-rich austenite. For austempering at 310 °C for 3 h in Figure 10b, 1.5 wt% of Ni had the most stable ausferrite, while for lower Ni alloying

ausferrite was over-austempered, having higher Voce parameters. For austempering at 250 °C for 6 h in Figure 10c, 1.5 wt% of Ni again had Voce parameters lower than the other compositions, attesting that 1.5 wt% of Ni ausferrite had the best stability. These conclusions were consistent with the conventional ductility analysis that can be used to support the optimal austempering time investigation in ADI production. In Figure 16, the mean elongations to rupture ($e_R$) for the different Ni contents and different austempering conditions are reported. For austempering at 380 °C for 1 h in Figure 16a, 0.0 wt% of Ni had the highest mean value $e_R$, while lower $e_R$ values are reported for ausferrite with higher Ni content. For 1.5 wt% Ni ADI, the austempering conditions were particularly under-austempered, causing an increase in Voce parameters and a dramatic reduction of ductility below 8%, as reported in Figure 16a, which explained why in Figure 1 the 1.5 wt% Ni ADI matched the minimum tensile properties complying with ISO 17804:2005 [48], while for the other compositions and austempering conditions ADIs had better tensile properties by far. So, an austempering time longer than 1 h could have improved the stability of ausferrite in 1.5 wt% Ni ADI, which is consistent with the fact that alloying makes the solid-state transformation more sluggish. For austempering at 310 °C for 3 h in Figure 16b, 1.5 wt% of Ni had the largest mean $e_R$, while at 250 °C for 6 h in Figure 16c, 1.5 wt% of Ni again had the best ductility, which was consistent with the MAD analysis.

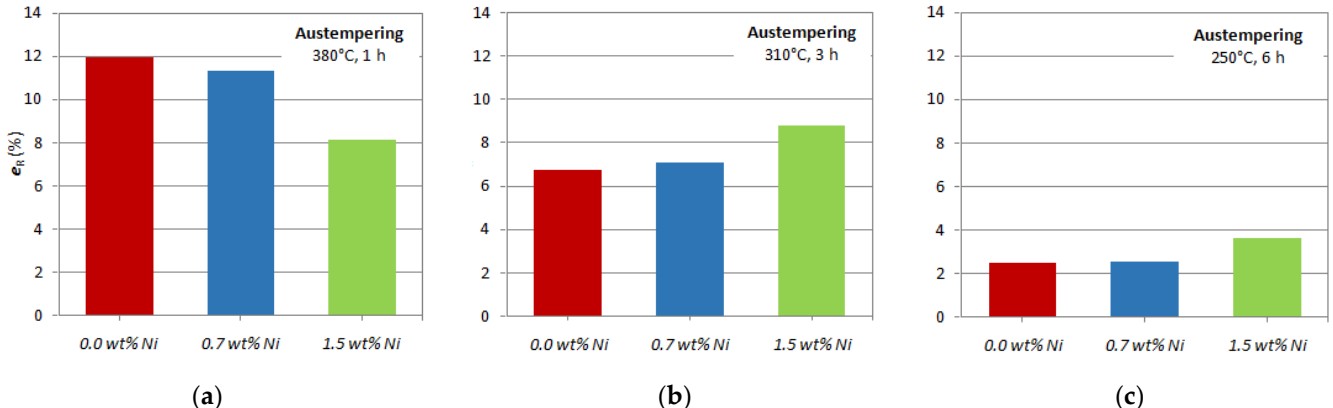

**Figure 16.** Mean elongations to rupture ($e_R$) from 25 mm Y-block ADIs for different Ni contents and different austempering conditions: (**a**) 380 °C for 1 h; (**b**) 310 °C for 3 h; (**c**) 250 °C for 6 h.

So, with increasing Voce parameter values that indicate an increase of ausferrite instability, there is a constant and consistent decrease of ductility, as summarized in Figure 17, where the correlation between ductility and Voce analysis strongly supports the new assessment approach of ausferrite stability based on MAD. However, the ductility itself can be misleading if carried out alone because of the wide variability typical of elongations to rupture, while the strain hardening behavior with the MAD approach is less sensitive to rupture and so can be used to validate and strengthen the conclusions about the goodness of the austempering conditions based on ductility. So, Voce analysis is a valid tool to support the ductility analysis for optimal determination of austempering conditions.

It is noteworthy that with increasing the Voce parameter values, their range width in MAD increased significantly, indicating that ausferrite stability and variability of tensile plastic behavior (Voce parameters variability) are correlated. So, in Figure 10, for instance, if the Voce parameter $1/\varepsilon_c$ is considered (the same results would be achieved considering $\Theta_o$, as they are linearly related) the best ausferrite stability was found for Ni 0.0 wt% austempered at 380 °C for 1 h with a mean value of $1/\varepsilon_c$ equal to 2.46: the biggest $1/\varepsilon_c$ value was 2.90, while the smallest $1/\varepsilon_c$ was 1.99. Conversely, the worst ausferrite stability was found for Ni 0.0 wt% austempered at 250 °C for 6 h with a mean value of $1/\varepsilon_c$ equal to 155.49: the biggest $1/\varepsilon_c$ value was 247.24, while the smallest $1/\varepsilon_c$ was 99.98. In Figure 18, the span width of the Voce parameter $1/\varepsilon_c$, that is, $\Delta 1/\varepsilon_c = (1/\varepsilon_c)_{max} - (1/\varepsilon_c)_{min}$, is reported vs. the mean value of $1/\varepsilon_c$ for any Ni content and austempering condition for the round

specimens from the 25 mm Y-block ADIs, showing that when increasing the ausferrite instability there is an increase of $\Delta 1/\varepsilon_c$, that is, an increase of plastic behavior variability.

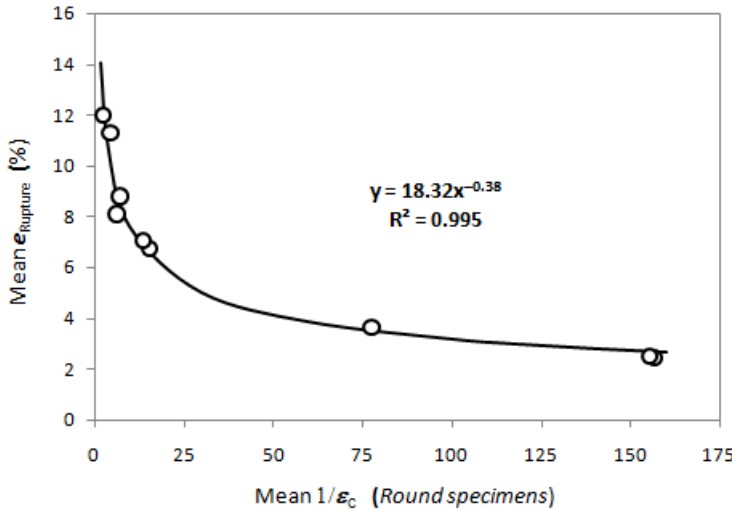

**Figure 17.** Mean $1/\varepsilon_c$ vs. mean elongation to rupture $e_{Rupture}$ (%) for any Ni content (0.0, 0.7 and 1.5 wt%) and austempering conditions (380 °C for 1 h; 310 for 3 h; 250 °C for 6 h) for the round specimens from the 25 mm Y-block ADIs.

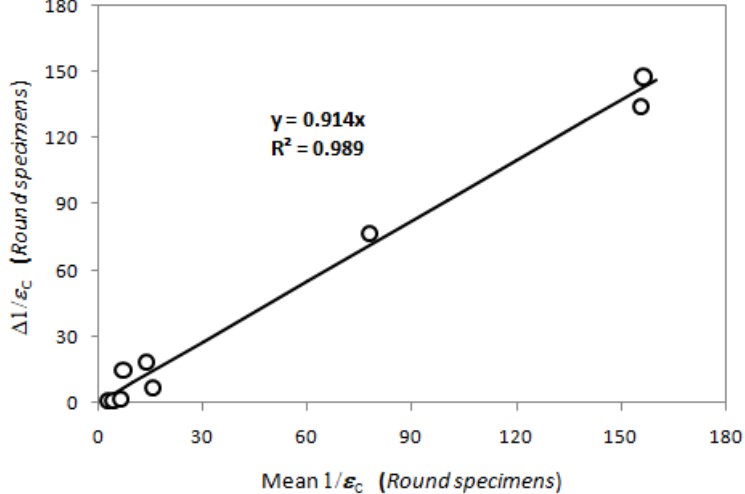

**Figure 18.** Span width $\Delta 1/\varepsilon_c = (1/\varepsilon_c)_{max} - (1/\varepsilon_c)_{min}$ vs. the mean value of $1/\varepsilon_c$ for any Ni content (0.0, 0.7 and 1.5 wt%) and austempering conditions (380 °C for 1 h; 310 °C for 3 h; 250 °C for 6 h) for the round specimens of the 25 mm Y-block ADIs.

In Figure 19, the volume fractions of metastable austenite are reported for different Ni contents and austempering conditions for the 25 mm Y-blocks. The ADIs with Ni alloying often presented the highest volume fractions of austenite at any austempering conditions, indicating that Ni fostered austenite retention, which is well known [39,41,42]. At 380 °C, the best ausferrite stability was for 0.0 wt% Ni content, even if the austenite volume fraction was not the highest if compared to 0.7 and 1.5 wt% of Ni ADIs, while the best ausferrite stability after austempering at 310 °C and 250 °C was for 1.5 wt% of Ni ausferrite, even if corresponding austenite volume fractions were not the highest. So, the volume fractions of austenite seemed to be not tightly related to the stability evaluation of ausferrite through MAD in Figure 12 or ductility in Figure 16.

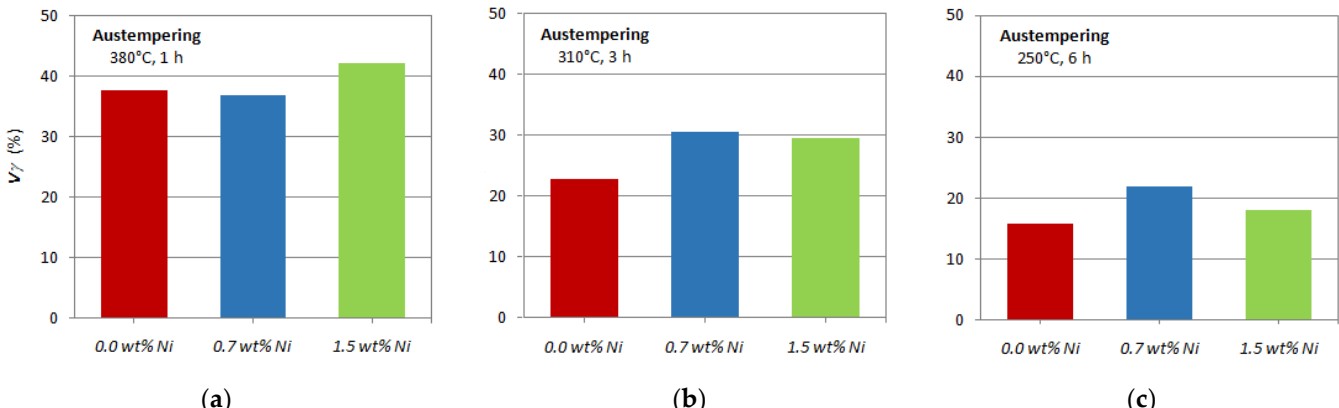

**Figure 19.** Volume fractions of austenite $V_\gamma$ in 25 mm Y-block ADIs for different Ni contents and different austempering conditions: (**a**) 380 °C for 1 h; (**b**) 310 for 3 h; (**c**) 250 °C for 6 h.

Indeed, what is relevant for ausferrite stability is to have the highest C content in austenite [1–6], which makes it stable at room temperature. In Figure 20, the C content values in austenite are reported for different Ni contents and austempering conditions from the 25 mm Y-blocks. In Figure 20, it is evident that the C content is more related to the austenite stability assessment through MAD in Figure 12 and to the ductility results in Figure 16. For instance, in Figure 20a, after austempering at 380 °C, the C content in austenite was the lowest in 1.5 wt% Ni ADI, consistent with the worst stability in MAD in Figure 12a, and the lowest ductility in Figure 16a (and the worst tensile properties in Figure 1). Again, in Figure 20b, after austempering at 310 °C, the C content in austenite was the lowest in 0.0 wt% Ni ADI for which the stability assessment in Figure 12b was the worst, and the ductility was the lowest in Figure 16b. The same results are for the austempering conditions 250 °C for 6 h in Figure 12c. Furthermore, in Figure 20c, the C content in austenite after austempering at 250 °C for 6 h is the lowest compared to the C content after austempering at 380 °C and 310 °C; this is consistent with the general trend found in Figure 12c, where the high values of Voce parameters indicated that the austenite stability was poor after austempering at 250 °C for any chemical composition, which was also consistent to the low ductility reported in Figure 16c. In conclusion, the C content in austenite was tightly consistent with the stability analysis through MADs in Figure 12 and the ductility results in Figure 16, while the austenite volume fraction was not.

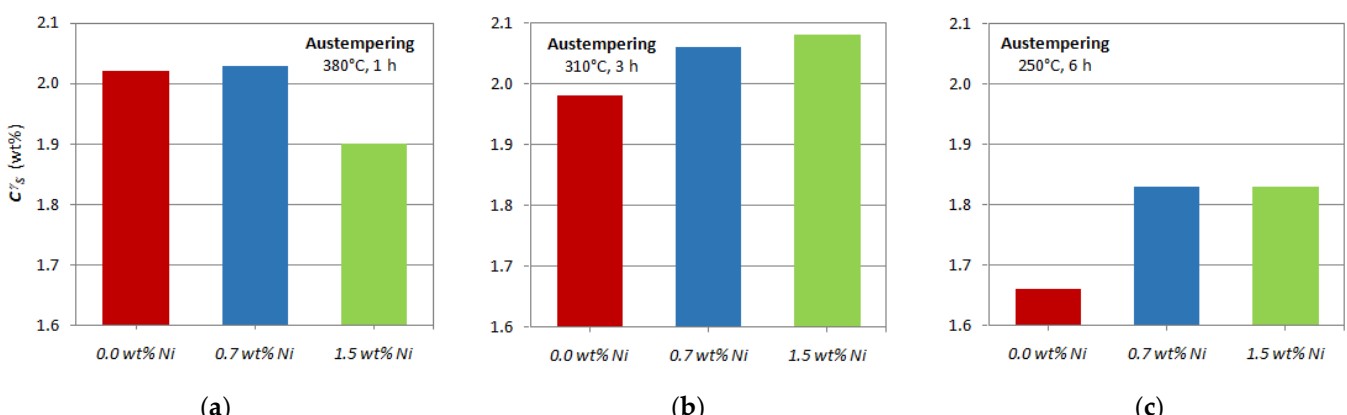

**Figure 20.** C content in austenite ($C^\gamma_S$) in 25 mm Y-block ADIs for different Ni contents and different austempering conditions: (**a**) 380 °C for 1 h; (**b**) 310 for 3 h; (**c**) 250 °C for 6 h.

*4.3. Tensile Behavior and Austenite Stability in ADIs Produced in Different Sections: 25 mm, 5 and 3 mm*

The MADs of the Voce data of the flat tensile specimens from 25 mm, 5 mm and 3 mm castings are reported in Figures 13–15 for different Ni contents and austempering conditions. In Figure 13, for the austempering at 380 °C for 1 h, the flat tensile specimen Voce data lie along the best fitting lines obtained by the round specimens data from 25 mm Y-blocks regardless of the Ni content, which indicated that ausferrite plastic behavior for different sections was the same for similar chemical compositions. However, the flat specimens data positions were higher than the round specimens data in MADs and random with no particular trend, suggesting that no ausferrite of the three sections (3 mm, 5 mm and 25 mm castings) had better ausferrite stability, while ausferrite instability increased. In fact, consistent with the findings on the correlation between Voce parameter variability and ausferrite instability in Figures 17 and 18, the ranges of Voce parameters of the flat tensile specimens were often wider than the ones from round tensile specimen data. Similar findings are reported in Figure 14 for austempering at 310 °C for 3 h: the flat tensile specimens data had random positions alongside the best linear fits of the round tensile specimens data regardless of the Ni content, and the ranges of Voce parameter values were wider than the data range from round specimens. In Figure 15, the instability of ausferrite after austempering at 250 °C for 6 h was quite high regardless of Ni content, section thickness and tensile specimen geometry, causing premature ruptures and low ductility; some data points are missing because the tensile flow curves did not have the sufficient plastic range for strain hardening analysis. This instability was consistent with the lower C content that was measured for all ADIs with different Ni contents after austempering at 250 °C for 6 h, reported in Figure 20c. Indeed, the data from 3 mm casting were the lowest for the three different Ni contents, which might suggest that the 3 mm casting ausferrite was more stable. However, the comparison of C contents in austenite in the 3, 5 and 25 mm castings in Figure 11 shows neither higher C content in austenite of the 3 mm castings nor higher volume fractions of austenite, and so no sure trend could be stated.

In Figures 13–15, the widths of the flat specimens Voce data ranges appeared to be wider than the Voce data ranges from round specimens for any Ni content and austempering conditions, which might have suggested that ausferrite stability could be lower in 5 mm and 3 mm castings rather than in 25 mm Y-blocks ADIs. However, the Voce parameters from flat tensile specimens of the 25 mm Y-block sections were also bigger, and the data ranges wider than the Voce data from the round tensile specimens from the same 25 mm Y-blocks, which indeed indicated that the tensile specimen geometry might have played some role in the instability of ausferrite. An example of flat and round tensile specimens' strain hardening behavior is reported in Figure 21 for 1.5 wt% Ni ADI from 25 mm Y-block austempered at 380 °C for 1 h.

The strain to rupture $\varepsilon_R$ from flat tensile specimens vs. the average strain to rupture from round tensile specimens from the same Ni contents and austempering conditions are reported in Figure 22a (flat tensile specimen data points of 0.0 wt% Ni ausferrite austempered at 310 °C for 3 h and at 250 °C for 6 h are missing, because the short plastic deformation ranges could not allow strain hardening analysis). In Figure 22a, data points are always below the dichotomy line where strains to rupture from flat and round tensile specimens should match if ductility were the same, regardless of Ni content and austempering conditions. This finding proved that ductility was always lower in flat tensile specimens than in round ones with the same ausferrite, resulting in an average ductility reduction of −21.6% with flat tensile specimens with respect to round tensile specimens. The fracture surfaces analysis did not reveal any difference between the fracture behaviors of round and flat tensile specimens (see Figure 7) that could help to rationalize the different ductility. When comparing the Voce analysis results on flat and round tensile specimens in Figure 22b, the parameters $1/\varepsilon_c$ from flat tensile specimens were always higher than the ones from round tensile specimens, with an average increase of 100.8%, indicating that the

ausferrite instability in flat tensile specimens was more significant, albeit the material was the same.

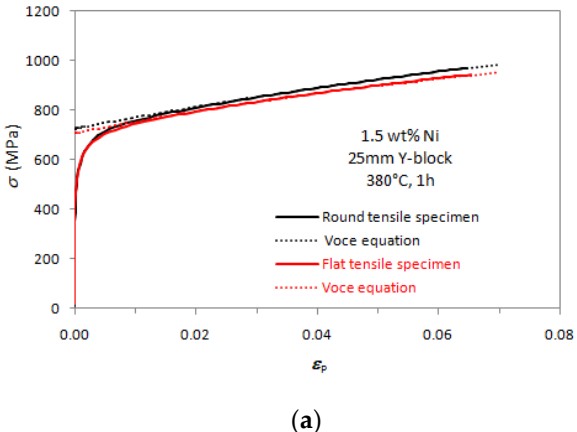
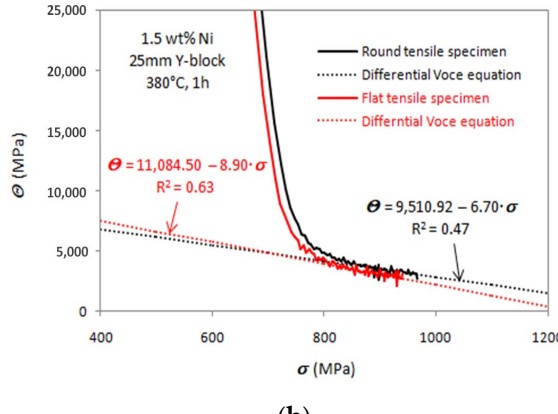

(**a**)

(**b**)

**Figure 21.** Comparison between tensile plastic behavior from round and flat tensile specimens: (**a**) tensile flow curves with round and flat tensile specimen of 1.5wt% Ni ADI from 25 mm Y-block austempered at 380 °C for 1 h; (**b**) differential data from the flow curves in (**a**) and strain hardening analysis based on differential Voce equation.

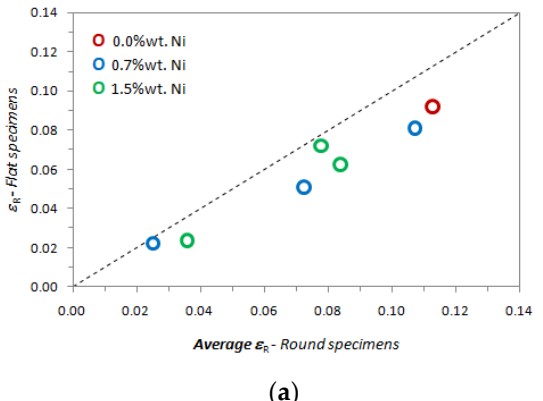
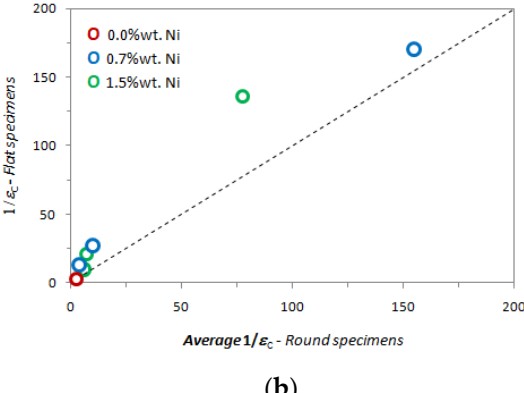

(**a**)

(**b**)

**Figure 22.** (**a**) Strains to rupture ($\varepsilon_R$) comparison from round and flat tensile specimens from 25 mm Y-block ADIs for different Ni contents and austempering conditions: $\varepsilon_R$—flat tensile specimens vs. average $\varepsilon_R$—round tensile specimens; (**b**) $1/\varepsilon_c$—flat tensile specimens vs. average $1/\varepsilon_c$—round tensile specimens (0.0 wt% Ni ausferrite austempered at 310 °C for 3 h and 250 °C for 6 h are missing).

Commercial purity polycrystalline Ni was tensile-tested with round and flat tensile specimens to assess whether the tensile specimen geometry could affect the tensile flow curves. Nickel engineering flow curves from round and flat tensile specimens are reported in Figure 23a for comparison. The flow curves matched up to the uniform elongation, that is, when the UTS was reached, and when localized deformation beyond uniform deformation occurred, there were significant deviations with major elongation reduction in the flat specimen flow curve. Different works [62,63] on the effects of flat tensile specimen geometry on mechanical properties have reported that the tensile flow curves were comparable to UTS as long as tensile specimen geometry complied with international standards. In [64], flow curves from round and flat specimens of a ferritic–pearlitic steel have reported the same results found in nickel in Figure 22, with good matching of the tensile flow curves up to UTS and significant deviation after localized deformation. With Weibull stress numerical calculation [64], the deviation between round and flat tensile specimens beyond uniformed elongations was rationalized to be caused by the different stress distributions at the localized-strain volumes beyond UTS.

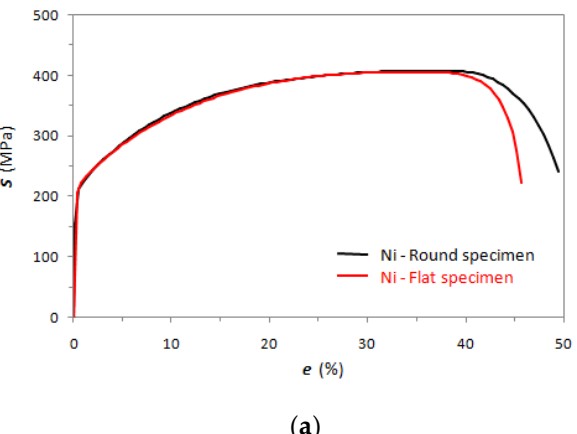
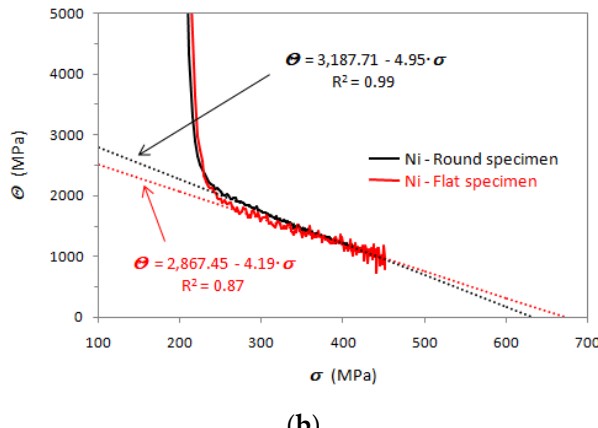

**(a)**　　　　　　　　　　　　　　　　　　　　**(b)**

**Figure 23.** Ni polycrystalline tensile plastic behavior comparison between round and flat tensile specimens data: (**a**) engineering stress–strain flow curves; (**b**) differential data (true strain hardening rate $\Theta$ vs. true stress $\sigma$) from the tensile flow curves in (**a**) and differential Voce equation fittings.

Because of the limited range of the extensometers, the flow curves in Figure 23a had to be stopped at the strain of about 17% and then set to zero for reloading again the specimens with strain control up to the final rupture. This procedure was not necessary with ADIs in the present investigation, as the ADIs' elongations to rupture were always below 14% (see Figure 1). In Figure 23b, differential data of the flow curves up to 0.017 in Figure 23a are reported. Indeed, the round and flat tensile specimen flow curves matched very well and the differential data almost superimposed up to about 17%, well over the typical elongations to rupture of the investigated ADIs, finding that $1/\varepsilon_c$ was 4.19 for flat tensile specimen data and 4.95 for round tensile specimen data. This indicated that the tensile specimen geometry did not affect the flow curves per se, and the effects found in ausferrite and reported in Figures 20 and 21 could be thus ascribed to ausferrite instability. Indeed, reduced ductility, increased Voce parameter values and Voce parameter variability (see Section 3.2) in the flow curves from flat tensile specimens showed that the specimen geometry played a role in the ADI tensile behavior, indicating that the flat geometry enhanced the ausferrite instability. To the authors' knowledge, this effect of tensile specimen geometry on ADIs has never been reported before.

In situ investigations on ADI plastic and fracture behavior have been reported [65–67], even if the main focus of these investigations was on the effects of graphite nodule cracking and graphite–matrix debonding on fracture evolution, rather than on ausferrite plastic behavior. Furthermore, these in situ investigations were only on flat small-sized tensile specimens not complying with international standards. In [67], it was reported that austenite caused some delays in the graphite–matrix debonding, while with bainitic ferrite, the debonding was easier and cracks propagated alongside the ferrite–austenite boundaries [65]. It could be speculated that these observations concerned the external surfaces of flat tensile specimens, causing ductility reduction, while in round tensile specimens, these mechanisms might be less active because of the reduction of the surface/volume ratio of the strain gauge. However, to the authors' knowledge, no insitu observations of ADI round tensile specimens have ever been reported, and further investigations will have to be carried out on this issue.

## 5. Conclusions

Ductile irons (DIs) with different nominal Ni contents were produced into 25 mm, 5 mm and 3 mm wall castings. The alloys were austenitized at the same conditions of 875 °C for 2 h and then austempered for three different combinations of temperatures and times: 250 °C for 6 h, 310 °C for 3 h and 380 °C for 1 h. So, the combined effects of section thickness, chemical composition and austempering conditions on the tensile mechanical properties of ausferrite were investigated and related to the ausferrite stability.

Strain hardening analysis of tensile flow curves was carried out to analyze the ADIs' tensile plastic behavior according to an innovative material quality assessment procedure based on strain hardening analysis and the Voce equation. Because of the different wall sections, round and flat tensile specimens with geometries complying with ASTM E8/E8M-11 [47] were tested. The following conclusions on the quality assessment procedure based on the Matrix Quality Assessment (MAD) could be drawn:

- MADs are tools for easy determination of the best austempering conditions: the Voce parameter positions in the diagrams indicate how stable the ausferrite is;
- MAD results are consistent with ductility analysis that is conventionally used to find the optimal austempering conditions; lower Voce parameters positions in MADs are consistent with higher ductility and better ausferrite stability, and vice versa: so, MAD and ductility analysis are suggested to be carried out in conjunction;
- In MADs, the widths of the Voce parameter range are also indications of ausferrite stability; high variability of Voce parameters means high instability of ausferrite.

About the effects of section thickness and Ni content on ausferrite in thin sections, the following conclusions could be drawn:

- Ni content promoted higher volume fractions of austenite in ADIs with similar austempering conditions;
- However, higher volume fractions of austenite did not necessarily mean a higher stability of ausferrite;
- Rather, a higher stability of ausferrite was related to higher content of C in metastable austenite, which was independent of Ni content;
- Though thinner sections produced better and finer graphite structure, ADIs from 25 mm, 5 mm and 3 mm castings with the same Ni content had ausferrite with similar tensile mechanical behavior, regardless of the thickness;
- Tensile specimen geometry affected the ausferrite stability, as flat tensile specimens enhanced the ausferrite instability compared to round tensile specimens;
- So, for proper analysis and comparison of the tensile properties of ADIs, the tensile specimen geometry has to be taken carefully into account.

**Author Contributions:** Conceptualization, G.A., M.G.; methodology, G.A.; investigation, G.A., R.D., D.R.; resources, G.A., B.C., M.G.; data analysis, G.A., M.G.; writing—original draft preparation, G.A.; writing—review and editing, R.D., D.R., F.B., B.C., M.G.; supervision, G.A. All authors have read and agreed to the published version of the manuscript.

**Funding:** This research received no external funding.

**Acknowledgments:** Davide Della Torre and Tullio Ranucci are warmly thanked for the technical work carried out in this investigation.

**Conflicts of Interest:** The authors declare that they have no conflict of interest.

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
