# Peer review of "On Ausferrite Produced in Thin Sections: Stability Assessment through Round and Flat Tensile Specimen Testing"

_metals, doi:10.3390/met13010105_

Round 1

Reviewer 1 Report

The topic of this article on the ausferrite produced in thin sections is interesting for me.

Many similar experimental data, e.g. Figures 11, 12, and 13 were shown in this paper and they were very difficult to read and realize for readers. Therefore, I suggust that the authors should organize and renew these data.

Many ASTM (E8/E8M-11, A536–84(2019)e1, ASTM E975-13, ASTM E2567-16a, ASTM A247-19 ) and ISO (ISO17804:2005) standards were not cited in this article. Because  the sources of these standards were important for this paper.

Based on above reasons, I reconsider this paper after major revision. 

Author Response

Dear Reviewer,

thanks a lot for the work you did to improve our article. All your suggestions were appreciated and the file of the list of the actions we took to reply to your comments and requirements, has been uploaded.

Kind regards

Giuliano Angella

Reviewer 2 Report

The manuscript submitted by Giuliano Angella represents a nice study on the ausferrite produced in thin sections. The study is well presented and give important conclusion for people working in the field. The introduction presents all important aspects of the problem, and the details of the experimental part are fully provided.

However, before publication, several amendments must be performed:

 1.      please add °C to all values in the abstract.

2.      Please provide a clear/sharpen image for Figure 2.

3.      The authors claim that they investigated the samples by secondary electrons and backscattered electrons, but in the manuscript are presented only some SEI images. Why?

4.      Again, about the microstructure: there is no image of microstructure at different Ni content.

5.      There is no image of the facture, regarding the section/flat round discussion.a

6.      Since the austenite faction is computed from X-ray diffraction pattern, at least for all 3 Ni content the diffraction peaks should be given.

7.      Line 592: “Because of the limited range of the extensometers, the flow curves in figure 21a had to be stopped at the strain of about 17%, and then set to zero for reloading again the specimens.” - It is correct, there will be the same result if the flow curve is recorded up to rupture?

Major revision is suggested.

Author Response

Dear Reviewer,

thanks a lot for your work and effort to improve our article. All your suggestions were appreciated and implemented in the text. The file of the list of the actions we took to reply to your comments and requirements, has been uploaded.

Kind regards

Giuliano Angella

.

Round 2

Reviewer 1 Report

I have already checked second review of this paper and the revised paper was written well. Therefore, I recommend ths article can be accepted in present form for publicaiton in Metals.   

Reviewer 2 Report

The authors have properly addressed all raised issues; therefore I recommend publication.